# Exposure to Benzo[a]pyrene and 1-Nitropyrene in Particulate Matter Increases Oxidative Stress in the Human Body

**DOI:** 10.3390/toxics11090797

**Published:** 2023-09-21

**Authors:** Sun-Haeng Choi, Bolormaa Ochirpurev, Akira Toriba, Jong-Uk Won, Heon Kim

**Affiliations:** 1Department of Occupational and Environmental Medicine, Chungbuk National University Hospital, Cheongju 28644, Republic of Korea; shchoi@cbnuh.or.kr; 2Department of Public Health, Graduate School, Yonsei University, Seoul 03722, Republic of Korea; juwon@yuhs.ac; 3Department of Preventive Medicine, College of Medicine, Chungbuk National University, Cheongju 28644, Republic of Korea; bolormaa@chungbuk.ac.kr; 4Department of Hygienic Chemistry, Graduate School of Biomedical Science, Nagasaki University, Nagasaki 852-8521, Japan; toriba@nagasaki-u.ac.jp

**Keywords:** polycyclic aromatic hydrocarbons, benzo[a]pyrene, 1-nitropyrene, oxidative stress, thiobarbituric acid-reactive substances, 8-hydroxydeoxyguanosine

## Abstract

Polycyclic aromatic hydrocarbons (PAHs) have been reported to cause oxidative stress in metabolic processes. This study aimed to evaluate the relationship between exposure to PAHs, including benzo[a]pyrene (BaP) and 1-nitropyrene (1-NP), in the atmosphere and oxidative stress levels in the human body. This study included 44 Korean adults who lived in Cheongju, Republic of Korea. Atmospheric BaP and 1-NP concentrations and urinary 6-hydroxy-1-nitropyrene (6-OHNP), N-acetyl-1-aminopyrene (1-NAAP), and 1-hydroxypyrene (1-OHP) concentrations were measured. The oxidative stress level was assessed by measuring urinary thiobarbituric acid-reactive substances (TBARS) and 8-hydroxydeoxyguanosine (8-OHdG) concentrations. Urinary TBARS and 6-OHNP concentrations significantly differed between winter and summer. BaP exposure was significantly associated with urinary 8-OHdG concentrations in summer. However, atmospheric 1-NP did not show a significant correlation with oxidative stress marker concentrations. Urinary 1-NAAP concentration was a significant determinant for urinary 8-OHdG concentration in summer. Oxidative stress in the body increases in proportion to inhalation exposure to BaP, and more 8-OHdG is produced in the body as the amount of 1-NP, which is metabolized to 1-AP or 1-NAAP, increases.

## 1. Introduction

Polycyclic aromatic hydrocarbons (PAHs) are organic compounds formed by the incomplete combustion or pyrolysis of organic substances, including tobacco and fossil fuels, and are composed of two or more fused benzene rings. Various types of PAHs exist in particulate matter (PM), a significant air pollutant.

Benzo[a]pyrene (BaP) is a representative PAH, and it has been reported that environmental or occupational exposure to BaP showed adverse effects on neurobehavioral function in adults and on neurodevelopment in children [1,2]. Moreover, the International Agency for Research on Cancer (IARC) classifies BaP as a human carcinogen [3]. BaP absorbed in the body induces cytochrome P450 1A1 activation and is metabolized to benzo[a]pyrene-7,8-diol-9,10-epoxide (BPDE) [4]. Excessive reactive oxygen species (ROS) are generated during this metabolic process [5]. 

Pyrene, one of the PAHs, is metabolized into 1-hydroxypyrene (1-OHP) by the cytochrome P450 enzyme and excreted into the urine. 1-OHP urinary concentration is frequently used as an indicator of exposure levels to PAHs [6]. 

Diesel exhaust particles (DEPs) are respirable particulate matter characterized by small particle size and large surface area [7]. This unique structure facilitates the adsorption of a number of substances, including PAHs and nitro-PAHs. Prolonged exposure to elevated levels of diesel exhaust (DE) has been linked to a range of detrimental health outcomes, including asthma, respiratory diseases, cardiovascular disorders, allergic sensitization, and even lung cancer [8,9,10]. In 2012, the International Agency for Research on Cancer classified DE as a Group 1 human carcinogen based on studies that found an increased risk of lung cancer associated with exposure to DE [9,11]. 

1-Nitropyrene (1-NP), a nitro-PAH, has been proposed as a marker for DE. Its presence in DE exceeds that of other particulate-bound nitro-PAHs and, importantly, its formation is not significantly driven by atmospheric photochemical reactions [12,13]. 1-NP has been reported to cause genetic mutations and chromosomal aberrations [14,15], and the IARC has classified 1-NP as a Group 2A chemical [11]. Exposure to 1-NP increases oxidative stress, inflammation, and endothelial dysfunction, resulting in an increased risk of cardiovascular disease. 

While it is recognized that no single analyte can perfectly represent the complex and variable composition of diesel particulate matter in all scenarios, the incorporation of 1-NP as a metric could potentially provide an accurate assessment of DE exposure. Furthermore, such an approach could also provide a more accurate representation of the carcinogenic nature of diesel exhaust [16]. Measurement of urinary concentrations of 1-NP metabolites, including 6-hydroxy-1-nitropyrene (6-OHNP), 1-aminopyrene (1-AP), and N-acetyl-1-aminopyrene (1-NAAP), is a valuable approach for assessing environmental exposure to 1-NP and DEPs.

The metabolic activation of PAHs leads to the production of reactive oxygen species (ROS), which play a crucial role in causing significantly elevated levels of oxidized guanine in nuclear DNA [17]. BaP and 1-NP are believed to cause oxidative stress and harmful effects on the human body during metabolism [5,14]; however, few studies have confirmed that ROS increases in the human body owing to BaP or 1-NP exposure.

PM concentration in Korea’s atmosphere is higher than those in developed countries, particularly in winter [18,19]. Cheongju, a city located in the center of Korea, has a large industrial complex and three waste incinerators; the PM brought across the border by the northwest wind in winter makes this city an area with a high PM concentration in Korea.

This study aimed to evaluate the relationship between exposure to PAHs, such as BaP and 1-NP, in the atmosphere and oxidative stress levels in the body of individuals living in Cheongju, Republic of Korea.

## 2. Materials and Methods

### 2.1. Study Participants

This study included 44 adults living around the industrial complex in Cheongju, a waste incinerator, or in suburban areas. Data on demographic factors, smoking habits, and daytime activity were collected using a questionnaire. All participants provided the first urine sample in the morning of the day following atmospheric sampling. Participants received a thorough explanation of the study and provided written consent.

This study was reviewed by the Institutional Review Board of Chungbuk National University (permission number: CBNU-201708-SDBR-0075).

### 2.2. BaP and 1-NP Atmospheric Air Concentration Measurement

Atmospheric air was sampled using a personal air sampler (Apex Standard, SN0376420 Casella CEL, Bedford, UK) once in summer and once in winter. A 37 mm PTFE filter (Teflo, Pall Corporation, Ann Arbor, MI, USA) with a pore size of 2 μm was connected to a personal air sampler, which was attached to the subject’s collar and sampled over 24 h at a rate of approximately 3 L/min.

After sampling, the filters were removed and placed in a flask, then 2 mL of dichloromethane was added. The flask was vortex mixed for 1 h and ultrasonically extracted for 30 min. The extract was completely dried and re-dissolved in 300 μL of acetonitrile. A one hundred-microliter aliquot of the solution was injected into a two-dimensional high-performance liquid chromatography (HPLC) system with a fluorescence detector (FD) for the quantification of BaP and 1-NP [20]. The injected sample was eluted through a clean-up column (Cosmosil, 5NPE, 150 × 4.6 mm i.d., 5 μm, Nacalai Tesque, Kyoto, Japan) with a guard column (10 × 4.6 mm i.d.); subsequently, 1-NP was reduced to 1-aminopyrene by using a reduction column (NPpak-RS, 10 × 4.6 mm i.d., Jasco, Tokyo, Japan) at 80 °C. The mobile phase used for the clean-up and reduction columns was ethanol/acetate buffer (pH, 5.5) (95/5, *v*/*v*) at a 0.2 mL/min flow rate. A fraction of 1-AP and unchanged BaP eluted from the reduction column with the mobile phase was mixed with 30 mM ascorbic acid at a 1.6 mL/min flow rate and subsequently trapped on the concentration column (Spheri-5 RP-18, 30 × 4.6 mm i.d., 5 μm, Perkin Elmer, MA, USA). The concentrated fraction was passed through two separation columns (Inertsil ODS-P, 250 × 4.6 mm i.d., 5 μm, GL Sciences, Tokyo, Japan) with their guard column (10 × 4.6 mm i.d.) in tandem. All columns except the reduction column were maintained at 20 °C. A programmed gradient elution of the separation columns was performed using 10 mM imidazole buffer (pH, 7.6) as eluent A and acetonitrile as eluent B. Finally, the separated analytes were detected using the dual-channel FD. The excitation/emission wavelengths were 260/420 and 254/425 nm for BaP and 1-AP, respectively. 

### 2.3. 8-Hydroxydeoxyguanosine (8-OHdG) Urinary Concentration Measurement

The 8-OHdG urinary concentration was measured using an ELISA kit (New 8-OHdG Check, JaICA, Fukuroi, Japan) according to the manufacturer’s instructions. A brief description of the measurement method is as follows: The urine samples and standards were added to an 8-OHdG-coated microtiter plate. After washing, an enzyme-labeled secondary antibody was added to the plate. Unbound HRP-conjugated secondary antibody was washed out, and the substrate solution was added. After the termination of the reaction, absorbance was read at 450 nm.

### 2.4. Thiobarbituric Acid-Reactive Substance (TBARS) Urinary Concentration Measurement

TBARS levels were measured at three different wavelengths (fluorescence, λ-ex 530 nm and λ-ex 550 nm; λ-ex 515 nm and λ-ex 553 nm; and absorbance, 532 nm) using a microplate reader [21]. The TBARS concentration was estimated using the following equation: TBARS level (μM) = −0.282 + 1.830 × (the TBARS level measured at the fluorescence wavelengths of λ-ex 530 nm and λ-em 550 nm, μM) −0.685 × (the TBARS level measured at the fluorescence wavelengths of λ-ex 515 nm and λ-em 553 nm, μM) + 0.035 × (the TBARS level measured at the absorbance wavelength of 532 nm, μM).

### 2.5. 1-OHP Urinary Concentration Measurement

To 45 mL of urine, 4.5 mL of 1-M sodium acetate buffer (pH, 5.0), 450 μL of β-glucuronidase/aryl sulfatase (type H-2, from *Helix pomatia*:  β-glucuronidase activity, 100,000 units/mL; and sulfatase activity, 7500 units/mL), and 50 mg of blue rayon were added, and the mixtures were incubated at 37 °C for 16 h. Blue rayon was removed, washed with water, air dried, and eluted with 2 mL of methanol and 2 mL of methanol + 1% NH_3_ solution. After evaporating to dry with N_2_ gas, it was dissolved again in 400 μL methanol. Fifty microliters of this eluent were used for 1-OHP concentration measurement, and the rest of the eluent was used for 6-OHNP and 1-NAAP concentration measurement.

The 1-OHP concentration was measured using an HPLC device (LC-20AD, Shimadzu, Kyoto, Japan) with a 250 × 4.6 mm reverse phase column (J’sphere ODS-H80, YMC, Kyoto, Japan) and a fluorescent detector (RF-20A, Shimadzu, Singapore). The mobile phase was 65% acetonitrile, and the flow rate was 1 mL/min. The excitation/emission wavelength of the FD was 242 nm/388 nm [22]. The limit of quantitation for the 1-OHP concentration measured by this method was 16.6 pg/L.

### 2.6. 6-OHNP and 1-NAAP Urinary Concentration Measurement

To assess the exposure level to 1-NP, we measured the concentration of 1-NP metabolites 6-hydroxy-1-nitropyrene (6-OHNP) and N-acetyl-1-aminopyrene (1-NAAP) in the urine. For the 6-OHNP and 1-NAAP concentration measurement, we used the remaining eluent used for the 1-OHP concentration measurement.

A nitro-reduction reaction was performed by passing the remaining eluent through the online reduction column (NPpak-RS). 6-Hydroxy-1-aminopyrene concentrations were measured with the same HPLC system with the same column and mobile phase as those used for the measurement of 1-OHP concentration. The excitation/emission wavelength of the FD was 285 nm/428 nm. The limit of quantitation for the 6-OHNP concentration measured by this method was 10.8 pg/L.

An amide reaction was performed at room temperature for 1 h by adding 200 μL of acetic anhydride to the eluent that had been nitro-reduced by passing it through an online reduction column. After extraction with 1 mL of dichloromethane, the extract was dried with N_2_ gas. After dissolving in 200 μL of 50% methanol, it was filtered through a 13 mm polytetrafluoroethylene syringe filter (PTFE-H, pore size of 0.2 μm, Hyundai Micro, Seoul, Republic of Korea) into a vial. Twenty microliters of the redissolved extract was injected into the same HPLC system with the same column and mobile phase as those used for the measurement of 1-OHP concentration. The excitation/emission wavelength of the FD was 273 nm/385 nm [13]. The limit of quantitation for the 6-OHNP concentration measured by this method was 4.6 pg/L. Since we used an amidification method of converting 1-AP to 1-NAAP, the 1-NAAP concentration measured in this study is the sum of the 1-AP and 1-NAAP concentrations. 

The urinary concentrations of those substances were corrected by the urinary creatinine concentration.

### 2.7. Statistical Analysis

Each variable was tested for normality, and paired t-tests were used for seasonal comparisons of normally distributed variables, and Wilcoxon signed-rank tests were used for seasonal comparisons of non-normally distributed variables. Regression analysis was performed to determine the relationship between BaP and 1-NP exposure levels, their urinary metabolite concentrations, and urinary 8-OHdG and TBARS concentrations. A general linear model was used to statistically analyze the effects of BaP or 1-NP exposure on 8-OHdG and TBARS urinary concentrations after controlling for age, sex, smoking habit, and levels of other exposure markers.

Statistical analysis was performed using SAS OnDemand (SAS Institute, Cary, NC, USA). A *p*-value of <0.05 was considered statistically significant.

## 3. Results

The general characteristics of the study participants are shown in Table 1. The mean age was 63.8 years, and approximately 60% (26/44) of participants were females. Current smokers accounted for 16.0% of participants. Residents living near the incinerator, those living near the industrial complex, and rural residents represented 36.4%, 22.7%, and 40.9% of the total number of participants, respectively.

BaP and 1-NP atmospheric air concentrations and TBARS, 8-OHdG, 1-OHP, 6-OHNP, and 1-NAAP urinary concentrations are presented in Table 2. The concentrations of BaP and 1-NP in the atmospheric air were significantly higher in winter than those in summer. TBARS, 6-OHNP, and 1-NAAP urinary concentrations were significantly different between winter and summer; however, 8-OHdG and 1-OHP urinary concentrations were not.

The results of the correlation analysis of PAHs for TBARS and 8-OHdG are shown in Figure 1 and Figure 2. BaP was significantly associated with 8-OHdG urinary concentration in summer. 1-NP atmospheric air concentration and 1-OHP and 6-OHNP urinary concentrations were not significant determinants for urinary TBARS and 8-OHdG concentrations. 1-NAAP urinary concentration showed a significant positive association with 8-OHdG urinary concentration in summer.

The results of the general linear model analysis are shown in Table 3. Current smoking status was associated with urinary TBARS and 8-OHdG concentrations in summer. Cumulative smoking amount was negatively associated with TBARS urinary concentration in summer. In the multivariate model, BaP was significantly associated with TBARS urinary concentration in winter and 8-OHdG urinary concentration in summer. 1-NP atmospheric air concentration and 1-OHP and 6-OHNP urinary concentrations were not significantly associated with TBARS and 8-OHdG urinary concentrations in any season. However, urinary 1-NAAP concentration was positively associated with 8-OHdG urinary concentration in both seasons.

## 4. Discussion

Atmospheric air concentrations of BaP and 1-NP were higher in winter than in summer. This is not only because more fossil fuels are burned for heating in winter but also because considerable amounts of PM generated in foreign countries are brought across the border by northwest winds. Nevertheless, the participants’ TBARS and 8-OHdG urinary concentrations were higher in summer than those in winter, and the TBARS urinary concentration was significantly different. Urinary concentrations of 6-OHNP were higher in winter, despite the fact that urine is more concentrated in summer than in winter, as the amount of 1-NP exposure is approximately three times higher in winter than in summer. However, urinary concentration of 1-NAAP, which is associated with oxidative stress, is 1.75 times higher in summer than in winter, which may explain the higher concentrations of oxidative stress markers excreted in the urine in summer than in winter. In addition, the fact that high temperatures and dehydration increase oxidative damage and induce antioxidant defenses in high-temperature environments may also explain the higher concentrations of oxidative stress markers in urine in summer than in winter [23].

BaP exposure was significantly associated with oxidative stress marker concentrations. However, the urinary concentration of 1-OHP, which is widely used as a biomarker for PAH exposure, did not show a significant correlation with TBARS and 8-OHdG urinary concentrations. Since 1-OHP is a pyrene metabolite, which is neither carcinogenic nor mutagenic, 1-OHP concentration does not show a significant relationship with these oxidative stress markers. As expected, since 1-OHP is not a metabolite of BaP, we found no significant correlation between participants’ atmospheric air BaP concentrations and 1-OHP urine concentrations (r = 0.059, *p* = 0.586). If the urinary concentration of BPDE, a specific exposure marker for BaP, is measured, a significant association is likely to be observed.

BaP is more likely to be carcinogenic in the presence of low-molecular-weight (LMW) PAHs. Addition of LMW PAHs to BaP-treated cells significantly increased micronucleus formation, gap junctional intercellular communication (GJIC) dysregulation, and cell cycle alterations compared to BaP alone. In addition, human lung cells treated with a binary mixture of B[a]P and LMW PAHs showed a significant increase in anti-BPDE–DNA adducts compared to cells treated with BaP alone, indicating that LMW PAHs have increased co-carcinogenic potential [24]. Romo et al. (2019) revealed that non-genotoxic, low-molecular-weight (LMW) PAHs trigger the activation of mitogen-activated protein kinase pathways and lead to the secretion of inflammatory mediators [25]. These processes play a role in the disruption of GJIC, which can be reversed by an anti-inflammatory inhibitor. In this study, we did not measure LMW PAH concentrations, so no results were available for the interaction of LMW PAHs and BaP. 

The 1-NP concentrations measured in this study (1.31 pg/m^3^ in winter and 0.44 pg/m^3^ in summer) are very low compared to occupational exposure levels stated in previous studies. The geometric mean of 1-NP concentrations ranged from 4.4 to 65 pg/m^3^ in a study by Riley et al. (2018) that examined 1-NP exposure levels in metal mine workers [26] and from 3 to 42,200 pg/m^3^ in a study by Scheepers et al. (2003) that examined 1-NP exposure levels in miners [27]. In our previous study of 70 South Koreans with occupational or environmental exposure to diesel exhaust, the mean 1-NP exposure level was 20.40 pg/m^3^ [28]. A study conducted in Kanazawa, Japan, found atmospheric 1-NP levels of 32 pg/m^3^ in 1994, a significantly higher concentration than that found in our study [29]. Interestingly, these levels have decreased over time and are now lower than average levels in Korea [30]. 1-NP levels in cities in China were found to be much higher than those in this study; the level of 1-NP in Shenyang, China, was approximately 80 pg/m^3^ in 2010 [31].

The level of exposure to 1-NP and urinary concentration of 6-OHNP, which is a 1-NP metabolite, did not show a significant correlation with urinary concentrations of the studied oxidative stress markers. However, urinary concentration of 1-NAAP, another 1-NP metabolite, was significantly associated with 8-OHdG urinary concentration. In the human body, absorbed 1-NP is mostly metabolized to 6-OHNP, 8-OHNP, or 3-OHNP, and less than 10%, on average, is metabolized to 1-AP or 1-NAAP [32,33]. These results suggest that 8-OHdG is produced when 1-NP is metabolized to 1-AP or 1-NAAP but not when 1-NP is metabolized to 6-OHNP [28]. 1-NP atmospheric air concentration showed a significant association with 6-OHNP urinary concentration (r = 0.291, *p* = 0.006) but not with that of 1-NAAP (r = −0.067, *p* = 0.534). Additionally, the relationship between 6-OHNP and 1-NAAP urinary concentrations was not significant (r = −0.109, *p* = 0.312). The proportion of 1-NP metabolized to 1-AP or 1-NAAP in the body is determined by the genetic polymorphism of the metabolic enzyme [28]. Therefore, in an individual, if the exposure level to 1-NP increases, urinary 1-AP or 1-NAAP concentration and, accordingly, 8-OHdG urinary concentration increase; however, in comparisons between individuals, a higher exposure level to 1-NP does not necessarily mean higher 1-AP, 1-NAAP, or 8-OHdG urinary concentrations. The 1-NAAP assay used in this study has a lower limit of detection than the 1-AP assay and is easier to measure because 1-AP is also converted to 1-NAAP and measured as 1-NAAP [28]. 

In this study, BaP exposure showed a significant positive correlation not only with urinary 8-OHdG but also with urinary TBARS. [34] BaP is metabolized by cytochrome P450 enzymes to various quinones which undergo one-electron reduction by microsomal NADPH–cytochrome P450 reductase, resulting in the formation of semiquinone anion radicals. During this metabolic transformation, the simultaneous generation of various ROS, especially hydroxyl radicals, may play a role in increasing the levels of 8-OHdG and MDA [35]. Although MDA and 8-OHdG are both biomarkers of oxidative stress, they reflect different aspects of oxidative damage. MDA is produced by lipid peroxidation, and high levels of MDA are associated with cardiovascular disease, inflammation, and oxidative stress-related diseases [36]. Since 8-OHdG is produced by ROS attacking DNA, 8-OHdG represents the level of oxidative stress affecting DNA [37]. Elevated levels of 8-OHdG in the body indicate increased oxidative damage to DNA, which is associated with several diseases, including aging, cancer, cardiovascular disease, and neurodegenerative disorders [34]. 8-OHdG also serves as an epigenetic modifier affecting transcriptional regulatory elements and other epigenetic modifications. Post-transcriptional regulation can also be redirected by 8-OHdG [38].

However, environmental air pollutants such as metals and PAHs might potentially prompt the generation of hydroxyl radicals via mechanisms like the Fenton reaction [39,40]. Subsequently, this can catalyze the formation of both 8-OHdG and MDA. The finding that urinary 1-NAAP levels were only statistically significant with urinary 8-OHdG levels suggests that there may be a slight difference between the adverse health effects that may be caused by BaP exposure and those caused by 1-NP exposure.

There have been several studies of increased levels of oxidative stress in cultured cells exposed to 1-NP [14,15] but no epidemiologic studies on the association between metabolite concentrations of 1-NP excreted in the urine and levels of oxidative stress. To our best knowledge, this is the first human study of the association between urinary concentrations of 1-NP metabolites and oxidative stress level. 

The first limitation of this study is the small sample size, but this study is not insignificant because it is the first epidemiologic study of this topic in humans. Another limitation of this study is that exposure levels to BaP and 1-NP were measured only twice, so it is less likely to be an accurate representation of average exposure levels over a long period of time. In general, inaccurate information on exposure weakens the power of statistical tests, leading to a toward-the-null effect, but in this study, there was a significant association between BaP exposure and oxidative stress and between urinary concentrations of 1-NP metabolites and oxidative stress. Therefore, this study is meaningful as a pilot study on the mechanisms of human toxicity of 1-NP.

In summary, current smoking status was associated with increased urinary concentrations of TBARS and 8-OHdG. Urinary concentrations of TBARS, which are associated with cardiovascular disease and inflammation, were also increased by environmental exposure to BaP. Even at low concentrations, as BaP inhalation exposure level and urinary 1-NAAP concentration increase, more 8-OHdG is produced in the body, which is associated with cancer, aging, cardiovascular disease, and neurodegenerative diseases.

## Figures and Tables

**Figure 1 toxics-11-00797-f001:**
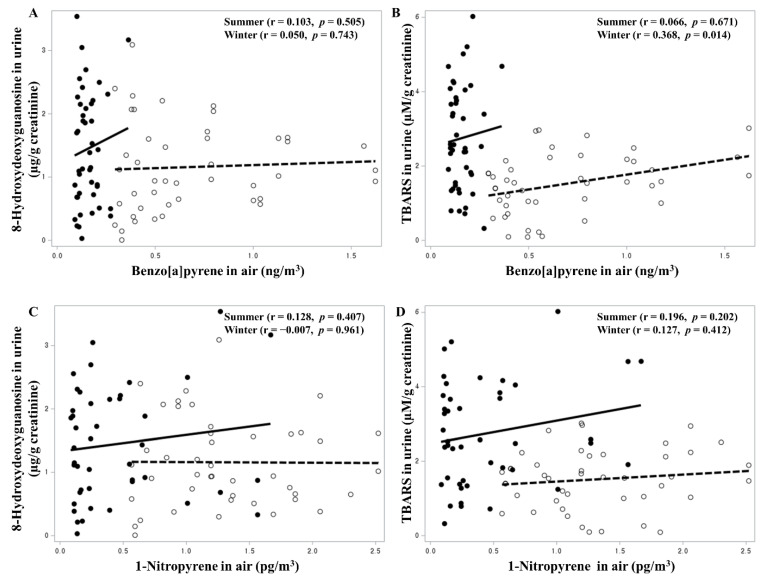
Correlation of atmospheric benzo[a]pyrene (**A**,**B**) and 1-nitropyrene (**C**,**D**) concentrations with urinary 8-hydroxydeoxyguanosine and TBARS concentrations (Winter: open circles and dashed lines. Summer: solid circles and solid lines).

**Figure 2 toxics-11-00797-f002:**
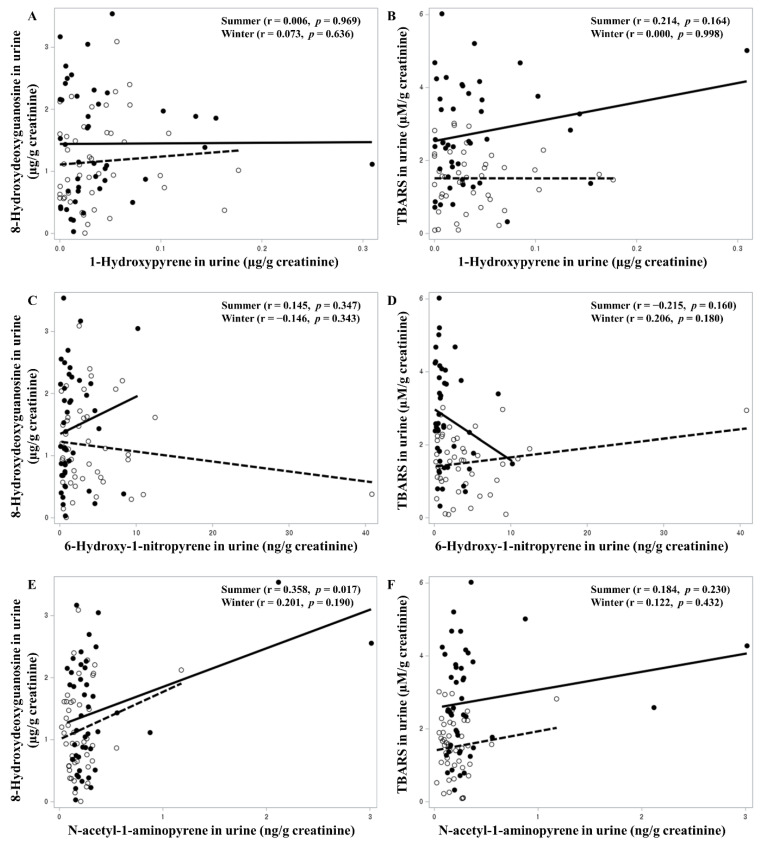
Correlation of urinary 1-hydroxypyrene (**A**,**B**), 6-hydroxy-1-nitropyrene (**C**,**D**), and N-acetyl-1-aminopyrene (**E**,**F**) concentrations with 8-hydroxydeoxyguanosine and TBARS concentrations (Winter: open circles and dashed lines. Summer: solid circles and solid lines).

**Table 1 toxics-11-00797-t001:** General characteristics of the study participants.

Variables		N (%)	Mean ± S.D.
Age		44 (100%)	63.8 ± 12.1 yr
Sex	Male	18 (40.9%)	
	Females	26 (59.1%)	
Smoking habit	Never smokers	28 (63.6%)	
	Ex-smokers	9 (20.5%)	
	Current smokers	7 (16.0%)	
Residential area	Incinerator area	16 (36.4%)	
	Industrial complex area	10 (22.7%)	
	Rural area	18 (40.9%)	

S.D.: standard deviation.

**Table 2 toxics-11-00797-t002:** Benzo[a]pyrene and 1-nitropyrene atmospheric air concentrations and TBARS, 8-hydroxydeoxyguanosine, 1-OHP, 6-OHNP, and 1-NAAP urinary concentrations.

Variables	Mean ± SD		*p*-Value
	Winter (N = 44)	Summer (N = 44)	
Benzo[a]pyrene in the atmospheric air (ng/m^3^)	0.68 ± 0.37	0.15 ± 0.06	<0.0001
1-Nitropyrene in the atmospheric air (pg/m^3^)	1.31 ± 0.54	0.44 ± 0.44	<0.0001
TBARS in the urine (μM/g creatinine)	1.51 ± 0.80	2.74 ± 1.39	<0.0001
8-Hydroxydeoxyguanosine in the urine (μg/g creatinine)	1.16 ± 0.70	1.44 ± 0.89	0.2142
1-OHP in the urine (μg/g creatinine)	0.04 ± 0.04	0.04 ± 0.06	0.5424
6-OHNP in the urine (ng/g creatinine)	4.30 ± 6.44	1.65 ± 2.15	0.0012
1-NAAP in the urine (ng/g creatinine)	0.20 ± 0.18	0.35 ± 0.52	0.0027

1-OHP, 1-hydroxypyrene; 6-OHNP, 6-hydroxy-1-nitropyrene; 1-NAAP, N-acetyl-1-aminopyrene.

**Table 3 toxics-11-00797-t003:** General linear model for TBARS and 8-hydroxydeoxyguanosine.

Variables	Season	TBARs in Urine (μM/g Creatinine)		8-OHdG in Urine (μg/g Creatinine)	
		β	*p*-Value	β	*p*-Value
Current smoker (no = 0, yes = 1)	Winter	−0.106	0.7962	−0.454	0.2485
	Summer	14.41	0.0412	0.990	0.0466
Cumulative smoking amount (Pack year)	Winter	0.011	0.2994	0.018	0.0748
	Summer	−0.052	0.0048	0.013	0.3089
Benzo[a]pyrene in air (ng/m^3^)	Winter	0.760	0.0251	0.207	0.8833
	Summer	5.566	0.1158	5.496	0.0313
1-Nitropyrene in air (pg/m^3^)	Winter	−0.271	0.2531	−0.156	0.5174
	Summer	0.345	0.4593	−0.309	0.3511
1-OHP in urine (μg/g creatinine)	Winter	−432.7	0.1662	276.6	0.3503
	Summer	407.0	0.4167	−657.5	0.0651
6-OHNP in urine (ng/g creatinine)	Winter	0.762	0.7038	−2.803	0.1482
	Summer	−11.22	0.2212	6.256	0.3324
1-NAAP in urine (ng/g creatinine)	Winter	49.08	0.4670	134.2	0.0373
	Summer	39.72	0.2899	55.69	0.0300

1-OHP, 1-hydroxypyrene; 6-OHNP, 6-hydroxy-1-nitropyrene; 1-NAAP, N-acetyl-1-nitropyrene.

## Data Availability

Data will be made available on request.

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
