# Peer review of "Exposure to Benzo[a]pyrene and 1-Nitropyrene in Particulate Matter Increases Oxidative Stress in the Human Body"

_toxics, 2023, doi:10.3390/toxics11090797_

Round 1

Reviewer 1 Report

The authors present results on the induction of oxidative stress by benzo[a]pyrene (BaP) and 1-nitropyrene (NP) in 44 adults living in Cheongju, Korea. Atmospheric concentrations of these two PAHs were measured and correlated with urinary levels for the metabolites of these two PAHs.  Oxidative stress was measured as a function of TBARS and 8-OHdG concentrations. Seasonal differences were noted. The following are suggested revisionary concerns:

1.      Written consent was given, however, the authors do not present sufficient information that the demographic data cannot be linked to identified individuals. The authors indicated the data is not publicly available but data is available upon request.  Please identify, or create a mechanism if not in place, how the demographic data has been or will be encrypted with protection to the identity of the individual subjects. I would assume the IRB would have made the same request.

2.      Please provide a summary cartoon of the results and potential for adverse health effects.

3.      Statistical analyses: Was a test of normality done. If not, please do so. If a test of normality fails, then use nonparametric analyses. I also recommend using ANOVA and do paired t-test only after the ANOVA null hypothesis is rejected.  Again, if normality failed, use the non-parametric Kruskal-Wallis one-way ANOVA on ranks before posthoc t-tests.

4.      The citation: “Jacobs et al. (2020) reported that under high temperature environments, high temperature and dehydration increase oxidative damage and induce anti-oxidant defense” is interesting but surely an overly simplistic explanation of the results of this study.

5.      Provide citations for the metabolism of pyrene into 1-OHP and for the common use of 1-OHP as a biomarker of PAH exposure.

6.      Also provide citations for 1-NP being specifically produced for diesel exhaust and citations that 1-NP causes genetic mutations and chromosomal aberrations.

7.      Also provide references for the statement that “BaP and 1-NP are believed to cause oxidative stress and harmful effects on the human body through these mechanisms.”

8.      Also provide references for the statement “PM concentration in Korea’s atmosphere is higher than those of in developed countries, particularly in winter.

9.      In general, please thoroughly review your paper for missing references such as lines 207 – 215 are missing several references.

10.   The statement “We observed no significant correlation between the BaP atmospheric air concentration and 1-OHP urinary concentration of the participants”.  This would be expected as 1-OHP is not a BaP metabolite, please indicate this and then segue into your next statement that BPDE, a specific metabolite of BaP did correlate.

11.   The authors are correct that MDA is a product of lipid peroxidation and 8OHdG is a product of DNA oxidation, but the present wording implies different oxidative mechanisms.  They both can result with the generation hydroxyl radicals through Fenton-like chemistries. How does this relate to their data as a function of PAH exposures. The BaP in particular is most associated with the oxidative stress markers.

12.   Reactive oxygen species are also involved in cell signaling events relevant to epigenetic effects, including 8-HOdG, which involves a more complicated picture to interpretations.1, 2 The implications of their results need to include potential epigenetic events. I recommend 2the use of these refs as well as others they will come across with an adequate literature search to provide a more robust involvement of oxidative stress in adverse health effects

13.   On the same note, the implications of PAHs in adverse health effects goes beyond genotoxicity and their results of 1-OHP is more relevant to these non-genotoxic effects. To emphasize only the potential adverse health outcomes to oxidative stress is incomplete. The following papers clearly indicate these non-genotoxic effects in cancer and inflammation and should be integrated into a discussion of their 1-OHP results as well as others the authors will find 3-11.

References

14.             1.            Hahm, J. Y.;  Park, J.;  Jang, E. S.; Chi, S. W., 8-Oxoguanine: from oxidative damage to epigenetic and epitranscriptional modification. Exp Mol Med 2022, 54 (10), 1626-1642.

15.             2.            Cooke, M. S.;  Evans, M. D.;  Dizdaroglu, M.; Lunec, J., Oxidative DNA damage: mechanisms, mutation, and disease. FASEB J 2003, 17 (10), 1195-214.

16.             3.            Bauer, A. K.;  Siegrist, K. J.;  Wolff, M.;  Nield, L.;  Bruning, T.;  Upham, B. L.;  Kafferlein, H. U.; Plottner, S., The Carcinogenic Properties of Overlooked yet Prevalent Polycyclic Aromatic Hydrocarbons in Human Lung Epithelial Cells. Toxics 2022, 10 (1).

17.             4.            Romo, D.;  Velmurugan, K.;  Upham, B. L.;  Dwyer-Nield, L. D.; Bauer, A. K., Dysregulation of Gap Junction Function and Cytokine Production in Response to Non-Genotoxic Polycyclic Aromatic Hydrocarbons in an In Vitro Lung Cell Model. Cancers (Basel) 2019, 11 (4).

18.             5.            Siegrist, K. J.;  Romo, D.;  Upham, B. L.;  Armstrong, M.;  Quinn, K.;  Vanderlinden, L.;  Osgood, R. S.;  Velmurugan, K.;  Elie, M.;  Manke, J.;  Reinhold, D.;  Reisdorph, N.;  Saba, L.; Bauer, A. K., Early Mechanistic Events Induced by Low Molecular Weight Polycyclic Aromatic Hydrocarbons in Mouse Lung Epithelial Cells: A Role for Eicosanoid Signaling. Toxicol Sci 2019, 169 (1), 180-193.

English language use was acceptable, but the manuscript lacked citations.

Author Response

The authors present results on the induction of oxidative stress by benzo[a]pyrene (BaP) and 1-nitropyrene (NP) in 44 adults living in Cheongju, Korea. Atmospheric concentrations of these two PAHs were measured and correlated with urinary levels for the metabolites of these two PAHs.  Oxidative stress was measured as a function of TBARS and 8-OHdG concentrations. Seasonal differences were noted. The following are suggested revisionary concerns:

  1. Written consent was given, however, the authors do not present sufficient information that the demographic data cannot be linked to identified individuals. The authors indicated the data is not publicly available but data is available upon request.  Please identify, or create a mechanism if not in place, how the demographic data has been or will be encrypted with protection to the identity of the individual subjects. I would assume the IRB would have made the same request.

The area where the subjects were recruited includes a village with a very small population, so it is possible to identify the subjects only by knowing their gender and age. That is why we did not disclose the data, but another reviewer is asking for disclosure, so we will submit the data with minimal personally identifiable information.

  1. Please provide a summary cartoon of the results and potential for adverse health effects.

As you pointed out, we have provided a summary cartoon of results:

  1. Statistical analyses: Was a test of normality done. If not, please do so. If a test of normality fails, then use nonparametric analyses. I also recommend using ANOVA and do paired t-test only after the ANOVA null hypothesis is rejected.  Again, if normality failed, use the non-parametric Kruskal-Wallis one-way ANOVA on ranks before posthoc t-tests.

As you pointed out, we modified our statistical analysis methods as follows.

“Each variable was tested for normality, and paired t-tests were used for seasonal comparisons of normally distributed variables, and Wilcoxon signed rank test were used for seasonal comparisons of non-normally distributed variables.”

  1. The citation: “Jacobs et al. (2020) reported that under high temperature environments, high temperature and dehydration increase oxidative damage and induce anti-oxidant defense” is interesting but surely an overly simplistic explanation of the results of this study.

We made the following improvements.

“Urinary concentration of 6-OHNP was higher in winter, despite the fact that urine is more concentrated in summer than in winter, as the amount of 1-NP exposure is approximately three times higher in winter than in summer. However, the urinary concentration of 1-NAAP, which is associated with oxidative stress, is 1.75 times higher in summer than in winter, which may explain the higher concentrations of oxidative stress markers excreted in the urine in summer than in winter. In addition, the fact that high temperatures and dehydration increase oxidative damage and induce antioxidant defenses in high temperature environments may also explain the higher concentrations of oxidative stress markers in urine in summer than in winter [24].”

  1. Provide citations for the metabolism of pyrene into 1-OHP and for the common use of 1-OHP as a biomarker of PAH exposure.

The following reference has been cited.

  1. Jongeneelen, F. J.; Bos, R. P.; Anzion, R. B.; Theuws, J. L.; Henderson, P. T. Biological monitoring of polycyclic aromatic hydrocarbons. Metabolites in urine.  Scand. J. Work. Environ. Health 198612(2), 137–143. https://doi.org/10.5271/sjweh.2166

  1. Also provide citations for 1-NP being specifically produced for diesel exhaust and citations that 1-NP causes genetic mutations and chromosomal aberrations.

The following references have been cited.

  1. Bamford, H. A.; Bezabeh, D. Z.; Schantz, S.; Wise, S. A.; Baker, J. E. Determination and comparison of nitrated-polycyclic aromatic hydrocarbons measured in air and diesel particulate reference materials. Chemosphere 200350(5), 575–587. https://doi.org/10.1016/s0045-6535(02)00667-7
  2. Toriba, A.; Kitaoka, H.; Dills, R. L.; Mizukami, S.; Tanabe, K.; Takeuchi, N.; Ueno, M.; Kameda, T.; Tang, N.; Hayakawa, K.; Simpson, C. D. Identification and quantification of 1-nitropyrene metabolites in human urine as a proposed biomarker for exposure to diesel exhaust. Chem. Res. Toxicol. 200720(7), 999–1007. https://doi.org/10.1021/tx700015q
  3. Kim, Y. D.; Ko, Y. J.; Kawamoto, T.; Kim, H. The effects of 1-nitropyrene on oxidative DNA damage and expression of DNA repair enzymes. J. Occup. Health 200547(3), 261–266. https://doi.org/10.1539/joh.47.261
  4. Andersson, H.; Piras, E.; Demma, J.; Hellman, B.; Brittebo, E. Low levels of the air pollutant 1-nitropyrene induce DNA damage, increased levels of reactive oxygen species and endoplasmic reticulum stress in human endothelial cells. Toxicol. 2009, 262(1), 57–64. https://doi.org/10.1016/j.tox.2009.05.008

  1. Also provide references for the statement that “BaP and 1-NP are believed to cause oxidative stress and harmful effects on the human body through these mechanisms.”

We modified the sentence to read as follows and cited the following references.

“The metabolic activation of PAHs leads to the production of reactive oxygen species (ROS), which play a crucial role in causing significantly elevated levels of oxidized guanine in nuclear DNA [18]. BaP and 1-NP are believed to cause oxidative stress and harmful effects on the human body during metabolism [5,14]; however, few studies have confirmed that ROS increases in the human body owing to BaP or 1-NP exposure.”

  1. Liu, C. S.; Tsai, C. S.; Kuo, C. L.; Chen, H. W.; Lii, C. K.; Ma, Y. S.; Wei, Y. H. Oxidative stress-related alteration of the copy number of mitochondrial DNA in human leukocytes. Free Radic. Res. 2003, 37(12), 1307-17. doi: 10.1080/10715760310001621342. PMID: 14753755.

  1. Yang, F.; Yang, H.; Ramesh, A.; Goodwin, J.S.; Okoro, E.U.; Guo, Z. Overexpression of Catalase Enhances Benzo(a)pyrene Detoxification in Endothelial Microsomes. PLoS One 2016, 11, e0162561.

  1. Kim, Y. D.; Ko, Y. J.; Kawamoto, T.; Kim, H. The effects of 1-nitropyrene on oxidative DNA damage and expression of DNA repair enzymes. J. Occup. Health 200547(3), 261–266. https://doi.org/10.1539/joh.47.261

  1. Also provide references for the statement “PM concentration in Korea’s atmosphere is higher than those of in developed countries, particularly in winter.

As you pointed out, we have provided references for the indicated sentence:

  1. Yun, S.-G.; Yoo, C. The Effects of spring and winter blocking on PM10 concentration in Korea. Atmosphere 2019, 10, 410. https://doi.org/10.3390/atmos10070410
  2. Allabakash, S.; Lim, S.; Chong, K.-S.; Yamada, T.J. Particulate Matter Concentrations over South Korea: Impact of Meteorology and Other Pollutants. Remote Sens. 2022, 14, 4849. https://doi.org/10.3390/rs14194849

  1. In general, please thoroughly review your paper for missing references such as lines 207 – 215 are missing several references.

As you pointed out, we have cited the following references.

  1. Ball, L.M.; Kohan, M.J.; Inmon, J.P.; Claxton, L.D.; Lewtas, J. Metabolism of 1-nitro[14C]pyrene in vivo in the rat and mutagenicity of urinary metabolites. Carcinogen. 1984, 5(12), 1557-64. doi: 10.1093/carcin/5.12.1557. PMID: 6499108.
  2. Ochirpurev, B.; Eom, S.Y.; Toriba, A.; Kim, Y.D.; Kim, H. Urinary 1-aminopyrene level in Koreans as a biomarker for the amount of exposure to atmospheric 1-nitropyrene. Toxicol. Res. 2021, 38(1), 45-51. doi: 10.1007/s43188-021-00096-z. PMID: 35070940; PMCID: PMC8748593.

  1. The statement “We observed no significant correlation between the BaP atmospheric air concentration and 1-OHP urinary concentration of the participants”.  This would be expected as 1-OHP is not a BaP metabolite, please indicate this and then segue into your next statement that BPDE, a specific metabolite of BaP did correlate.

We modified that sentence to read as follows

“As expected, since 1-OHP is not a metabolite of BaP, we found no significant correlation between participants' BaP air concentrations and 1-OHP urine concentrations (r = 0.059, p = 0.586).”

  1. The authors are correct that MDA is a product of lipid peroxidation and 8OHdG is a product of DNA oxidation, but the present wording implies different oxidative mechanisms.  They both can result with the generation hydroxyl radicals through Fenton-like chemistries. How does this relate to their data as a function of PAH exposures. The BaP in particular is most associated with the oxidative stress markers.

Thank you for your insightful comment regarding the potential implication of oxidative mechanisms for MDA and 8-OHdG. We appreciate the clarification on the shared potential for the generation of hydroxyl radicals through Fenton-like chemistries for both markers. To address your point more explicitly, we will rephrase our statement to avoid implying distinct oxidative mechanisms for MDA and 8OHdG. Thank you again for bringing this important nuance to our attention. We believe that addressing it will provide readers with a clearer understanding of the relationship between BaP exposure and oxidative stress markers. The revised sentence in the discussion section of the revised manuscript is as follows.

“In this study, BaP exposure showed a significant positive correlation not only with urinary 8-OHdG but also with urinary TBARS. [35] BaP is metabolized by cytochrome P450 enzymes to various quinones which undergo one-electron reduction by microsomal NADPH-cytochrome P450 reductase, resulting in the formation of semiquinone anion radicals. During this metabolic transformation, the simultaneous generation of various ROS, especially hydroxyl radicals, may play a role in increasing the levels of 8-OHdG and MDA [36]. Although MDA and 8-OHdG are both biomarkers of oxidative stress, they reflect different aspects of oxidative damage. MDA is produced by lipid peroxidation, and high levels of MDA are associated with cardiovascular disease, inflammation, and oxidative stress-related diseases [37]. Since 8-OHdG is produced by ROS attacking DNA, 8-OHdG represents the level of oxidative stress on DNA [38]. Elevated levels of 8-OHdG in the body indicate increased oxidative damage to DNA, which is associated with several diseases, including aging, cancer, cardiovascular disease, and neurodegenerative disorders [35].”

  1. Reactive oxygen species are also involved in cell signaling events relevant to epigenetic effects, including 8-HOdG, which involves a more complicated picture to interpretations.1, 2The implications of their results need to include potential epigenetic events. I recommend 2the use of these refs as well as others they will come across with an adequate literature search to provide a more robust involvement of oxidative stress in adverse health effects

As pointed out by the reviewer, we have added the following sentence to the discussion.

“8-OHdG also serves as an epigenetic modifier affecting transcriptional regulatory elements and other epigenetic modifications. Post-transcriptional regulation can also be redirected by 8-OHdG [39].”

  1. On the same note, the implications of PAHs in adverse health effects goes beyond genotoxicity and their results of 1-OHP is more relevant to these non-genotoxic effects. To emphasize only the potential adverse health outcomes to oxidative stress is incomplete. The following papers clearly indicate these non-genotoxic effects in cancer and inflammation and should be integrated into a discussion of their 1-OHP results as well as others the authors will find 3-11.
  2. Hahm, J. Y.;  Park, J.;  Jang, E. S.; Chi, S. W., 8-Oxoguanine: from oxidative damage to epigenetic and epitranscriptional modification. Exp Mol Med 2022,54 (10), 1626-1642.
  3. Cooke, M. S.;  Evans, M. D.;  Dizdaroglu, M.; Lunec, J., Oxidative DNA damage: mechanisms, mutation, and disease. FASEB J 2003,17 (10), 1195-214.
  4. Bauer, A. K.;  Siegrist, K. J.;  Wolff, M.;  Nield, L.;  Bruning, T.;  Upham, B. L.;  Kafferlein, H. U.; Plottner, S., The Carcinogenic Properties of Overlooked yet Prevalent Polycyclic Aromatic Hydrocarbons in Human Lung Epithelial Cells. Toxics 2022,10 (1).
  5. Romo, D.;  Velmurugan, K.;  Upham, B. L.;  Dwyer-Nield, L. D.; Bauer, A. K., Dysregulation of Gap Junction Function and Cytokine Production in Response to Non-Genotoxic Polycyclic Aromatic Hydrocarbons in an In Vitro Lung Cell Model. Cancers (Basel) 2019,11 (4).
  6. Siegrist, K. J.;  Romo, D.;  Upham, B. L.;  Armstrong, M.;  Quinn, K.;  Vanderlinden, L.;  Osgood, R. S.;  Velmurugan, K.;  Elie, M.;  Manke, J.;  Reinhold, D.;  Reisdorph, N.;  Saba, L.; Bauer, A. K., Early Mechanistic Events Induced by Low Molecular Weight Polycyclic Aromatic Hydrocarbons in Mouse Lung Epithelial Cells: A Role for Eicosanoid Signaling. Toxicol Sci 2019,169 (1), 180-193.

As pointed out by the reviewer, we have added the following sentences to the discussion.

“BaP is more likely to be carcinogenic in the presence of low molecular weight (LMW) PAHs. Addition of LMW PAHs to BaP-treated cells significantly increased micronucleus formation, gap junctional intercellular communication (GJIC) dysregulation, and cell cycle alterations compared to BaP alone. In addition, human lung cells treated with a binary mixture of B[a]P and LMW PAHs showed a significant increase in anti-BPDE-DNA adducts compared to cells treated with BaP alone, indicating that LMW PAHs have increased co-carcinogenic potential [25]. Romo et al. (2019) revealed that non-genotoxic, low molecular weight (LMW) PAHs trigger the activation of mitogen-activated protein kinase pathways and lead to the secretion of inflammatory mediators [26]. These processes play a role in the disruption of GJIC, which can be reversed by an anti-inflammatory inhibitor. In this study, we did not measure LMW PAH concentrations, so no results were available for the interaction of LMW PAHs and BaP.”

Reviewer 2 Report

In the manuscript (toxics-2542108), entitled “Exposure to benzo[a]pyrene and 1-nitropyrene in particulate matter increases oxidative stress in the human body” atmospheric concentration of benzo[a]pyrene (BaP) and 1-nitropyrene (1-NP) were measured and its oxidative stress effect to human body was studied. This study is not meet the requirements of the TOXIXS in its current form.  The related points are listed below. 

1) Please present the ethical community number for the research permission.  

2) Brand of PTFE filter should be presented.  In addition, brands and models of the instruments should be presented.  

3) The sampling method for PM is not clear. Is the equipment attached to patients or placed in a special area? Please defined the conditions.

4) Please revised the following sentences.

Line 76: The filter was placed in a flask, 2 mL of dichloromethane was added, vortex mixed, and ultrasonically extracted. 

5)   The vortex and ultrasound times should be presented. 

6) How many mL of acetonitrile was used for the re-dissolving step? 

7) How many mL of sample was injected to HPLC? 

8) Please define the procedure step by step by describing each method separately between Lines 80-91. 

9) Please placed “ 2.3.8. -Hydroxydeoxyguanosine (8-OHdG) Urinary Concentration Measurement” on Page 3. 

10) Line 112 and Line 126: Please correct “–OHP…” as “1–OHP…” and “-OHNP” as “6-OHNP”. Please check the title numbers. 

11) Number of repetitions for the analysis should be presented. I don’t mean the injection repetition (sampling and analysis).

12) What are the quantification limits for the targeted compounds? 

13) Please rearrange Table 3 by describing β. Because some result looks like presenting the concentration and the concentration cannot have a negative value. 

14) The data used for the statistical analysis should be presented as a supplementary file. 

15) Line 167: Please write “f” in the “figure 2” by a capital letter. 

16) "5. Conclusions" part should be improved. 

17) Reference style should be controlled. 

18) The result should be compared with the other published articles. 

Author Response

In the manuscript (toxics-2542108), entitled “Exposure to benzo[a]pyrene and 1-nitropyrene in particulate matter increases oxidative stress in the human body” atmospheric concentration of benzo[a]pyrene (BaP) and 1-nitropyrene (1-NP) were measured and its oxidative stress effect to human body was studied. This study is not meet the requirements of the TOXIXS in its current form.  The related points are listed below. 

1) Please present the ethical community number for the research permission.  

We added the ethical community number (CBNU-201708-SDBR-0075).

2) Brand of PTFE filter should be presented.  In addition, brands and models of the instruments should be presented.  

The brand name of the PTFE filter was Teflo from Pall Corporation.

We have added the brand name as follows;

“A 37-mm PTFE filter (Teflo, Pall Corporation, Ann Arbor, MI) with a pore size of 2 μm was connected to a personal air sampler, which was attached to the subject's collar and sampled over 24 hours at a rate of approximately 3 L/min.”

3) The sampling method for PM is not clear. Is the equipment attached to patients or placed in a special area? Please defined the conditions.

Subjects were asked to wear the sampler with the filter holder attached to their collar.

We have changed the sentence as follows;

“A 37-mm PTFE filter (Teflo, Pall Corporation, Ann Arbor, MI) with a pore size of 2 μm was connected to a personal air sampler, which was attached to the subject's collar and sampled over 24 hours at a rate of approximately 3 L/min.”

4) Please revised the following sentences.

Line 76: The filter was placed in a flask, 2 mL of dichloromethane was added, vortex mixed, and ultrasonically extracted. 

We have fixed that sentence as shown below.

“After sampling, the filters were removed and placed in a flask, 2 mL of dichloromethane was added. The flask was vortex mixed for 1 hr, and ultrasonically extracted for 30 min.”

5)   The vortex and ultrasound times should be presented. 

We vortex mixed for 1 hr and ultrasonically extracted for 30 min.

6) How many mL of acetonitrile was used for the re-dissolving step? 

The extract was re-dissolved in 300 mL of acetonitrile.

7) How many mL of sample was injected to HPLC? 

Aliquots of the 100 mL solution were injected into the HPLC system.

8) Please define the procedure step by step by describing each method separately between Lines 80-91. 

We have changed that paragraph as the reviewer pointed out.

9) Please placed “ 2.3.8. -Hydroxydeoxyguanosine (8-OHdG) Urinary Concentration Measurement” on Page 3. 

We have fixed the phrase as follows; 2.3. 8-Hydroxydeoxyguanosine (8-OHdG) Urinary Concentration Measurement”

10) Line 112 and Line 126: Please correct “–OHP…” as “1–OHP…” and “-OHNP” as “6-OHNP”. Please check the title numbers. 

We have fixed the title number as you point out.

11) Number of repetitions for the analysis should be presented. I don’t mean the injection repetition (sampling and analysis).

All analyses except BaP, 1-NP, and 6-OHNP were performed in duplicate.

12) What are the quantification limits for the targeted compounds? 

The limits of quantitation not shown elsewhere are as follows;

 1-OHP: 16.6 pg/L

6-OHNP: 10.8 pg/L

1-NAAP: 4.6 pg/L

We have included them in the text.  

13) Please rearrange Table 3 by describing β. Because some result looks like presenting the concentration and the concentration cannot have a negative value. 

We have corrected Table 3. 

14) The data used for the statistical analysis should be presented as a supplementary file. 

We have submitted the data as a supplementary file.

15) Line 167: Please write “f” in the “figure 2” by a capital letter. 

We've made the changes as you instructed.

16) "5. Conclusions" part should be improved. 

The conclusion has been edited as follows;

“In summary, current smoking was associated with increased urinary concentrations of TBARS and 8-OHdG. Urinary concentrations of TBARS, which are associated with cardiovascular disease and inflammation, were also increased by environmental exposure to BaP. Even at low concentrations, as BaP inhalation exposure level and urinary 1-NAAP concentration increase, more 8-OHdG is produced in the body, which is associated with cancer, aging, cardiovascular disease, and neurodegenerative diseases.”

17) Reference style should be controlled. 

We've fixed it as pointed out by the reviewer.  

18) The result should be compared with the other published articles. 

As the reviewer pointed out, we compared the results of this study with those of other studies.

“The 1-NP concentrations measured in this study (1.31 pg/m3 in winter and 0.44 pg/m3 in summer) are very low compared to the occupational exposure levels in previous studies. The geometric mean of 1-NP concentrations ranged from 4.4 to 65 pg/m3 in a study by Riley et al. (2018) that examined 1-NP exposure levels in metal mine workers [27], and from 3 to 42200 pg/m3 in a study by Scheepers et al. (2003) that examined 1-NP exposure levels in miners [28]. In our previous study of 70 South Koreans with occupational or environmental exposure to diesel exhaust, the mean 1-NP exposure level was 20.40 pg/m3 [29]. A study conducted in Kanazawa, Japan found atmospheric 1-NP levels of 32 pg/m3 in 1994, a significantly higher concentration than that found in our study [30]. Interestingly, these levels have decreased over time and are now lower than average levels in Korea [31]. 1-NP levels in cities in China were much higher than those in this study. The level of 1-NP in Shenyang, China was approximately 80 pg/m3 in 2010 [32].”

Reviewer 3 Report

The manuscript titled “Exposure to benzo[a]pyrene and 1-nitropyrene in particulate matter increases oxidative stress in the human body” the authors evaluate the relationship between exposure to PAHs, in- cluding benzo[a]pyrene (BaP) and 1-nitropyrene (1-NP), in the atmosphere and oxidative stress levels in the human body. The paper is interesting and the topic is suitable for the Toxics journal, furthermore the PAHs are widely spread pollutants and very dangerous for ecosystems and human health. Despite this, I believe that the study has not been carried out correctly, in fact the Introduction Section is not very informative and contains few innovative news, the study was carried out only on 44 adults, the discussions need to be expanded, the conclusions are extremely brief, the bibliographic supports are only 12. In this state the study cannot be published. My opinion is to reject the paper.

Author Response

We have significantly enhanced the Introduction, Discussion, and Conclusion sections, and discussed the limitation of this study. The number of references has been increased to 41.

And even though there were only 44 subjects, the number of measurements was 88, as PAH exposure levels and urinary concentrations were measured in summer and winter, respectively.    

Since this is the first human study of the association between urinary concentrations of 1-NP metabolites and oxidative stress level, we believe this study is meaningful as a pilot study on the mechanism of human toxicity of 1-NP.

Round 2

Reviewer 1 Report

Authors adequately addressed my concerns.

Reviewer 2 Report

In the manuscript "toxics-2542108" entitled " entitled "Exposure to benzo[a]pyrene and 1-nitropyrene in particulate matter increases oxidative stress in the human body"  required changes have been done and it can be acceptable in its current version.